# The mRNA decapping machinery targets *LBD3/ASL9* to mediate apical hook and lateral root development

Zhangli Zuo[1], Milena E Roux[1], Jonathan R Chevalier[1], Yasin F Dagdas[2], Takafumi Yamashino[3], Søren D Højgaard[1], Emilie Knight[4], Lars Østergaard[4], Eleazar Rodriguez[1], Morten Petersen[1]

Multicellular organisms perceive and transduce multiple cues to optimize development. Key transcription factors drive developmental changes, but RNA processing also contributes to tissue development. Here, we report that multiple decapping deficient mutants share developmental defects in apical hook, primary and lateral root growth. More specifically, *LATERAL ORGAN BOUNDARIES DOMAIN 3 (LBD3)/ASYMMETRIC LEAVES 2-LIKE 9 (ASL9)* transcripts accumulate in decapping deficient plants and can be found in complexes with decapping components. Accumulation of *ASL9* inhibits apical hook and lateral root formation. Interestingly, exogenous auxin application restores lateral roots formation in both *ASL9* over-expressors and mRNA decay–deficient mutants. Likewise, mutations in the cytokinin transcription factors type-B ARABIDOPSIS RESPONSE REGULATORS (B-ARRs) *ARR10* and *ARR12* restore the developmental defects caused by over-accumulation of capped *ASL9* transcript upon *ASL9* overexpression. Most importantly, loss-of-function of *asl9* partially restores apical hook and lateral root formation in both *dcp5-1* and *pat* triple decapping deficient mutants. Thus, the mRNA decay machinery directly targets *ASL9* transcripts for decay, possibly to interfere with cytokinin/auxin responses, during development.

## Introduction

Understanding proper tissue development requires information about diverse cellular mechanisms controlling gene expression. Much work has focused on the transcriptional networks that govern stem cell differentiation. For example, ectopic expression of *LATERAL ORGAN BOUNDARIES DOMAIN (LBD)/ASYMMETRIC LEAVES 2-LIKE (ASL)* genes is sufficient to induce spontaneous proliferation of pluripotent cell masses in plants, a reprogramming process triggered in vitro by complementary/Yin-Yang phytohormones auxin

and cytokinin (Fan et al, 2012; Schaller et al, 2015). Auxin and cytokinin responses are essential for a vast number of developmental processes in plants including postembryonic reprograming and formation of the apical hook to protect the meristem during germination in darkness (Chaudhury et al, 1993; Hu et al, 2017) and lateral root (LR) formation (Jing & Strader, 2019). Loss-of-function mutants in genes that regulate auxin-dependent transcription such as *auxin-resistant1 (axr1)* exhibit defective hooking and LR formation (Estelle & Somerville, 1987; Lehman et al, 1996). In addition, type-B ARABIDOPSIS RESPONSE REGULATORS (B-ARRs) ARR1, ARR10, and ARR12 work redundantly as transcriptional activators to regulate cytokinin targets including type-A ARRs, which are negative regulators of cytokinin signaling in shoot development and LR formation (Riefler et al, 2006; Ishida et al, 2008; Xie et al, 2018). Exogenous cytokinin application disrupts LR initiation by blocking pericycle founder cell transition from G2 to M phase (Li et al, 2006; Laplaze et al, 2007). Thus, reshaping the levels of certain transcription factors leads to changes in cellular identity. As developmental programming must be tightly regulated to prevent spurious development, the expression of these transcription factors may be controlled at multiple levels (Tatapudy et al, 2017). However, most developmental studies focus on their transcription rates and overlook the contribution of mRNA stability or decay to these events (Crisp et al, 2016).

Eukaryotic mRNAs contain stability determinants including the 5′ 7-methylguanosine triphosphate cap (m7G) and the 3′ poly-(A) tail. mRNA decay is initiated by deadenylation, followed by degradation via either 3′–5′ exosomal exonucleases and SUPPRESSOR OF VCS (SOV)/DIS3L2 or via the 5′–3′ exoribonuclease (XRN) activity of the decapping complex (Garneau et al, 2007; Sorenson et al, 2018). This complex includes the decapping holoenzyme composed of the catalytic subunit Decapping 2 (DCP2) and its cofactor DCP1 along with other factors (DCP5, DHH1, VCS, LSM1-7 complex, and PAT1), and the XRN that degrades monophosphorylated mRNA. As a central platform, PAT1 (Protein Associated with Topoisomerase II, PAT1b in mammals) forms a heterooctameric complex with LSM (Like-sm)1–7 at 3′ end of a

[1]Department of Biology, Faculty of Science, University of Copenhagen, Copenhagen, Denmark  [2]Gregor Mendel Institute, Austrian Academy of Sciences, Vienna BioCenter, Vienna, Austria  [3]Laboratory of Molecular Microbiology, School of Agriculture, Nagoya University, Nagoya, Japan  [4]Crop Genetics Department, John Innes Centre, Norwich Research Park, Norwich, UK

Correspondence: shutko@bio.ku.dk

mRNA to engage transcripts containing deadenylated tails thereafter recruits other decapping factors and interacts with them using different regions; these decapping complex and mRNAs can aggregate into distinct cytoplasmic foci called processing bodies (PBs) (Brengues et al, 2005; Balagopal & Parker, 2009; Ozgur et al, 2010; Chowdhury et al, 2014; Charenton et al, 2017; Lobel et al, 2019). Beyond *DCP* genes, deletion of *PAT1* gene in yeast exhibits the strongest temperature sensitive phenotype compared with other decapping factors genes (Bonnerot et al, 2000).

mRNA decay regulates mRNA levels and thereby impacts cellular reprogramming (Newman et al, 2017; Essig et al, 2018). We and others have shown that the decapping machinery is involved in stress and immune responses (Xu & Chua, 2012; Merret et al, 2013; Roux et al, 2015; Perea-Resa et al, 2016; Crisp et al, 2017; Yu et al, 2019), and that RNA-binding proteins can target selected mRNAs for decay (Gerstberger et al, 2014; Perea-Resa et al, 2016; Yu et al, 2019). Postembryonic lethality (Xu et al, 2006) and stunted growth phenotypes (Xu & Chua, 2009; Perea-Resa et al, 2012) associated with disturbance of the decay machinery indicate the importance of mRNA decapping and decay machinery during plant development. However, although much has been learned about how mRNA decapping regulates plant stress responses (Perea-Resa et al, 2016; Yu et al, 2019; Zuo et al, 2021), far less is known about how decapping contributes to plant development.

*Arabidopsis dcp1*, *dcp2*, and *vcs* mutants display postembryonic lethality, whereas *lsm1alsm1b*, *pat* triple mutant, and *dcp5* knock-down mutants only exhibit abnormal development (Xu et al, 2006; Xu & Chua, 2009; Perea-Resa et al, 2012; Zuo et al, 2022a, 2022b *Preprint*). All these differences suggest that mutations in mRNA decay components may cause pleiotropic phenotypes not directly linked to mRNA decapping and decay deficiencies (Riehs-Kearnan et al, 2012; Gloggnitzer et al, 2014; Roux et al, 2015). For example, it has been proposed that lethality in some mRNA decay loss-of-function mutants is not due to decay deficiencies per se but to the activation of immune receptors which evolved to surveil microbial manipulation of the decay machinery (Roux et al, 2015). In line with this, loss-of-function of *AtPAT1* inappropriately triggers the immune receptor SUMM2, and *Atpat1* mutants consequently exhibit dwarfism and autoimmunity (Petersen et al, 2000; Zhang et al, 2012; Roux et al, 2015; Rodriguez et al, 2020). Thus, PAT1 is under immune surveillance and PAT proteins are best studied in SUMM2 loss-of-function backgrounds.

Here, we studied the impact of mRNA decapping during development. For this, we have analyzed three sequential mRNA decapping mutants *dcp2-1*, *dcp5-1*, and *pat* triple mutant (*pat1-1path1-4path2-1summ2-8*), revealing that the mRNA decay machinery targets the important developmental regulator *ASL9*. Specifically, disruption of the mRNA decay machinery promotes *ASL9* accumulation, and this in turn contributes to inhibit apical hook and lateral root formation. Interestingly, these developmental defects, which are observed in mRNA decapping deficient mutants and *ASL9* over-expressors, can be salvaged through disruption of cytokinin signaling or exogenous application of auxin. Importantly, mutations in *asl9* also partially restores the developmental defects including apical hook and lateral root formation in decapping mutants. These observations indicate that the mRNA decay machinery is fundamental to developmental decision-making.

# Results

## mRNA decapping deficiency causes deregulation of apical hooking

We and others have reported that mutants of mRNA decay components exhibit abnormal developmental phenotypes including postembryonic death and stunted growth (Xu et al, 2006; Xu & Chua, 2009; Perea-Resa et al, 2012; Roux et al, 2015; Zuo et al, 2022b *Preprint*), indicating mRNA decay may be needed for proper development. To assess this, we explored readily scorable phenotypic evidence of defective development. Because apical hooking can be exaggeratedly induced by exogenous application of ethylene or its precursor ACC, we germinated seedlings in darkness in the presence or absence of ACC (Bleecker et al, 1988; Guzman & Ecker, 1990). Interestingly, all three sequential mRNA decapping mutants tested *dcp2-1*, *dcp5-1*, and *pat* triple mutant were hookless and unable to make the exaggerated apical hook under ACC treatment (Figs 1A and B and S1A and B), being that *dcp2-1* exhibit the strongest hookless phenotype. Because *dcp2-1* is postembryonic lethal, we used seeds from a parental heterozygote to score for hook formation, and subsequentially confirmed by genotyping that all hookless seedlings were *dcp2-1* homozygotes. This, and the fact that ACC treatment leads to massive increase of DCP5–GFP (Chicois et al, 2018) and Venus–PAT1(Zuo et al, 2022b *Preprint*) foci in hook regions (Fig 1C), all suggest that mRNA decapping is required for apical hooking.

## mRNA decay machinery targets *ASL9* for decay

To search for transcripts responsible for the hookless phenotype, we revisited our previous RNA-seq data for *pat* triple mutant (Zuo et al, 2022b *Preprint*) and verified that transcripts of *ASL9* (*ASYMMETRIC LEAVES 2-LIKE 9*, also named *LBD3*, *LOB DOMAIN-CONTAINING PROTEIN 3*) accumulated specifically in *pat* triple mutants (Zuo et al, 2022b *Preprint*). ASL9 belongs to the large AS2/LOB (ASYMMETRIC LEAVES 2/LATERAL ORGAN BOUNDARIES) family (Matsumura et al, 2009) which includes key regulators of organ development (Xu et al, 2016). Interestingly, the ASL9 homologue ASL4 negatively regulates brassinosteroids accumulation to limit growth in organ boundaries, and overexpression of *ASL4* impairs apical hook formation and leads to dwarfed growth (Bell et al, 2012). Although *ASL4* mRNA did not accumulate in *pat* triple mutants (Zuo et al, 2022b *Preprint*), we hypothesized that ASL9 could also interfere with apical hook formation. We therefore analyzed mRNA levels of *ASL9* in ACC-treated seedlings and verified that all three sequential mRNA decapping mutants accumulated up to 30-fold higher levels of *ASL9* transcript compared with ACC-treated Col-0 seedlings (Fig 2A). Concordantly, two over-expressor lines of *ASL9* Col-0/*oxASL9* and Col-0/*oxASL9-VP16* (Naito et al, 2007) also exhibited hookless phenotypes (Fig 2B and C). However, we did not observe any changes including tighter apical hooks in *asl9-1* mutants (Fig S1C and D), suggesting other members of the AS2/LOB family act redundantly in this process. Nevertheless, these results indicate that apical hook formation in mRNA decapping deficient mutants is compromised, in part, might be due to misregulation of *ASL9*.

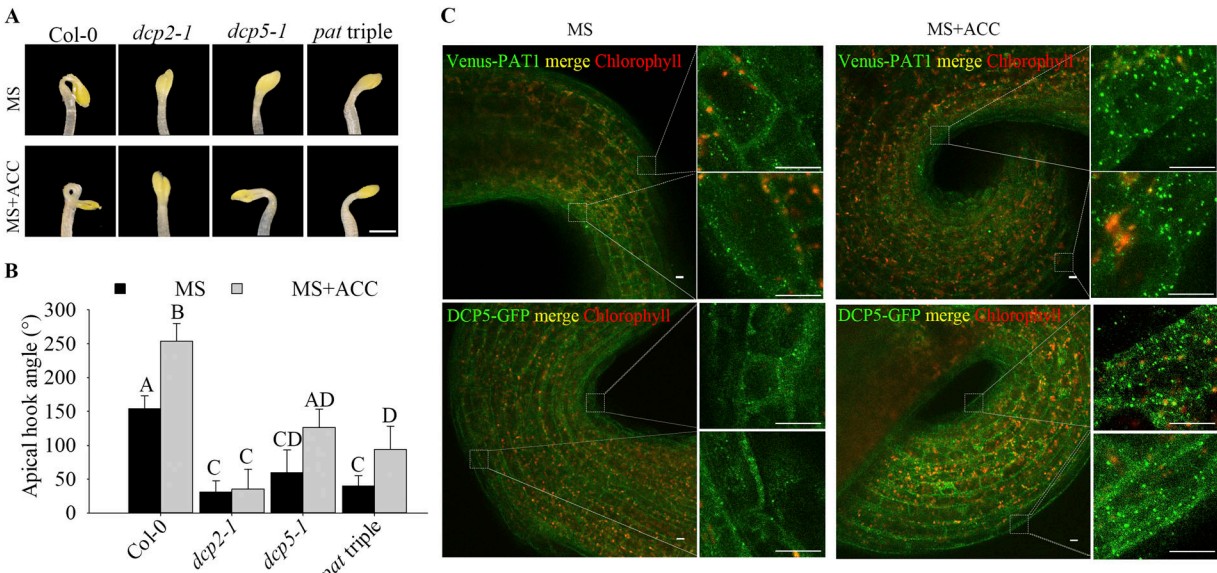

**Figure 1. mRNA decapping deficiency causes deregulation of apical hooking.**
**(A, B)** Hook phenotypes (A) and apical hook angles (B) in triple response to ACC treatment of etiolated Col-0, *dcp2-1*, *dcp5-1*, and *pat* triple seedlings. The experiment was repeated three times, in each repeat sample size (n)>30 for each genotype and treatment, and representative pictures are shown. The scale bar indicates 1 mm. Bars marked with the same letter are not significantly different from each other (*P*-value > 0.05). **(C)** Representative confocal microscopy pictures of hook regions following ACC treatment. Dark-grown seedlings with either Venus–PAT1 (top) or DCP5–GFP (bottom) on MS or MS + ACC plates for 4 d. Scale bars indicate 10 *μ*m.

To determine whether *ASL9* is a target of the decapping complex, we performed 5′-RACE assays and found significantly higher levels of capped *ASL9* in mRNA decapping mutant seedlings than in Col-0 (Fig 2D). We also assayed for capped *ASL9* transcripts in ACC and mock-treated mRNA decapping mutants. By calculating the ratio between capped and total *ASL9* transcripts, we verified that with ACC treatment, mRNA decapping mutants accumulated significantly higher levels of capped *ASL9* transcripts than Col-0 (Fig 2E). Moreover, RNA immunoprecipitation (RIP) revealed enrichment of *ASL9* in DCP5–GFP and Venus–PAT1 plants compared with a MYC–YFP control line (YFP-WAVE) (Fig 2F), indicating mRNA-decapping components directly bind *ASL9* transcripts. These data confirm that *ASL9* mRNA can be found in mRNA-decapping complexes, and that mRNA decapping regulates *ASL9* mRNA levels and contributes to ACC-induced apical hook formation.

### Accumulation of *ASL9* suppresses LR formation

LR formation is another example of postembryonic development. In *Arabidopsis*, the first stage of LR formation requires that xylem pericycle pole cells change fate to become LR founder cells, a process positively regulated by auxin and negatively regulated by cytokinin and ethylene (Jung & McCouch, 2013; Weijers et al, 2018). We therefore examined LR formation in mRNA decapping deficient mutants *dcp5-1* and *pat* triple mutants and in both *ASL9* over-expressors and verified that LR formation was dramatically impaired in all genotypes tested (Figs 3A and B and S2A and B). However, like seen for apical hooking, *asl9-1* also appeared to display normal LR formation (Fig S2C and D). Nevertheless, LR formation defects in *dcp5-1* and *pat* triple mutants indicate that mRNA decapping is required for the commitment to LR formation.

This is further substantiated by the fact that auxin application leads to a massive increase of DCP5–GFP and Venus–PAT1 foci in root regions (Fig 3C). Collectively, these data indicate mRNA decapping machinery, targeting *ASL9*, also contributes to LR formation.

### *ASL9* contributes to apical hooking and LR formation

The overexpression of *ASL9* is sufficient to suppress apical hook and lateral root development. To examine more directly if *ASL9* accumulation contributes to the developmental defects in decapping mutants, we crossed *asl9-1* to both *dcp5-1* and *pat* triple mutant to generate *dcp5-1asl9-1* and *pats asl9-1* (*pat1-1path1-4path2-1summ2-8asl9-1*) mutants. We then germinated *dcp5-1asl9-1* and *pats asl9-1* seedlings in darkness in the presence or absence of ACC, and under both conditions, *dcp5-1asl9-1* and/or *pats asl9-1* made more stringent hooks than *dcp5-1* and/or *pat* triple but not as tight as Col-0 or *asl9*-1 did, indicating that the loss-of-function of *asl9* can partially suppress decapping deficient mutants hookless phenotype (Figs 4A and B and S3A and B). Moreover, the LR phenotype of *dcp5-1* and *pat* triple was also partially restored by mutating *ASL9* (Figs 4C and D and S3C and D). Thus, our data indicate that *ASL9* contributes to both apical hooking and LR development in mutant with decapping deficiencies.

### Interference with cytokinin signaling and/or exogenous auxin restores developmental defects of *ASL9* over-expressor and mRNA decay–deficient mutants

*ASL9* has been implicated in cytokinin signaling (Naito et al, 2007; Ye et al, 2021) in which ARR1, ARR10, and ARR12 are responsible for activation of cytokinin transcriptional responses (Ishida et al, 2008; Xie et al, 2018), and cytokinin acts antagonistically with auxin. Apical

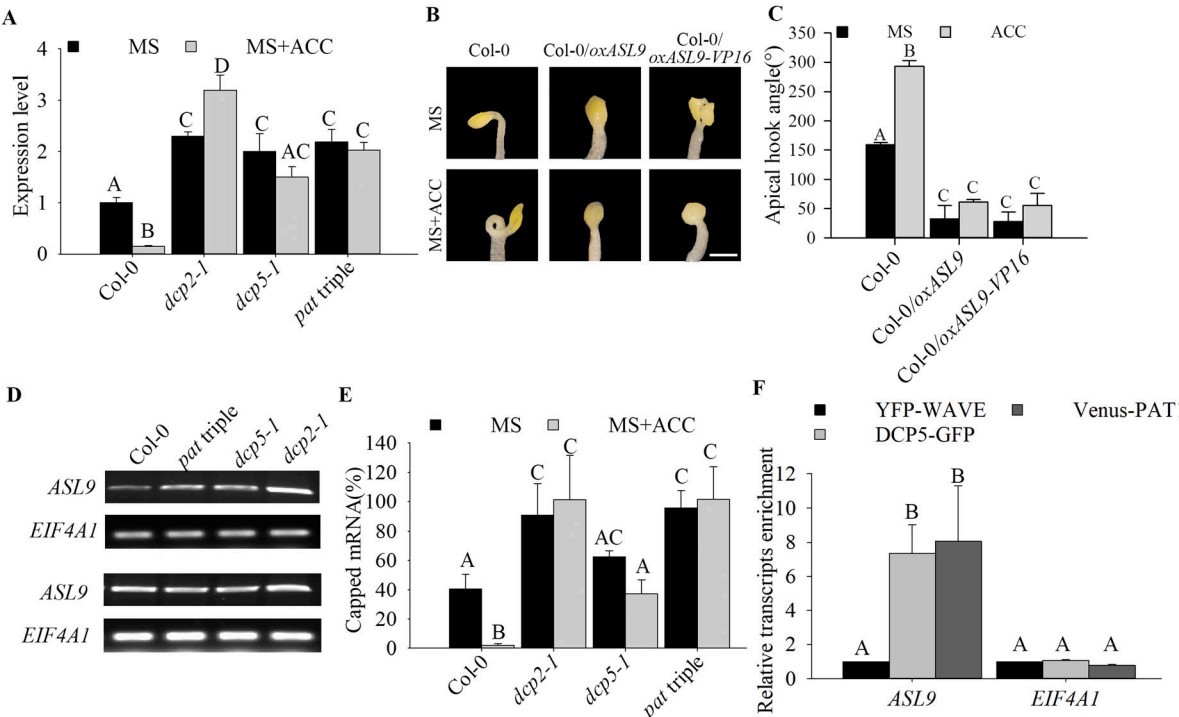

**Figure 2. mRNA decay machinery targets *ASL9* for decay.**
**(A)** *ASL9* mRNA levels in cotyledons and hook regions of dark-grown Col-0, *dcp2-1*, *dcp5-1*, and *pat* triple seedlings under control or ACC treatment. Error bars indicate SE of bio-triplicates. **(B, C)** Hook phenotypes (B) and apical hook angles (C) of triple response to ACC treatment of etiolated seedlings of Col-0, Col-0/*oxASL9*, and Col-0/*oxASL9-VP16*. The experiment was repeated three times, in each repeat sample size (n)>15 for each genotype and treatment, and representative pictures are shown. The scale bar indicates 1 mm. **(D)** Accumulation of capped transcripts of *ASL9* analyzed in 4-d-old MS grown etiolated seedlings of Col-0, *pat* triple, *dcp5-1*, and *dcp2-1* by 5′-RACE-PCR. RACE-PCR products obtained using low (upper panel) and high (bottom panel) number of cycles are shown. *EIF4A1* RACE-PCR products were used as loading control. **(E)** Capped *ASL9* transcript levels using XRN1 susceptibility assay in cotyledons and hook regions of dark-grown Col-0, *dcp2-1*, *dcp5-1*, and *pat* triple seedlings. Error bars indicate SE (n = 3). **(F)** DCP5 and PAT1 bind *ASL9* transcripts. 4-d dark-grown plate seedlings with DCP5–GFP or Venus–PAT1were taken for RIP assay. *ASL9* transcript levels were normalized to those in RIP of YFP-WAVE as a non-binding control. *EIF4A1* was used as a negative control. Error bars indicate SE (n = 3). Bars marked with the same letter are not significantly different from each other (*P*-value > 0.05).

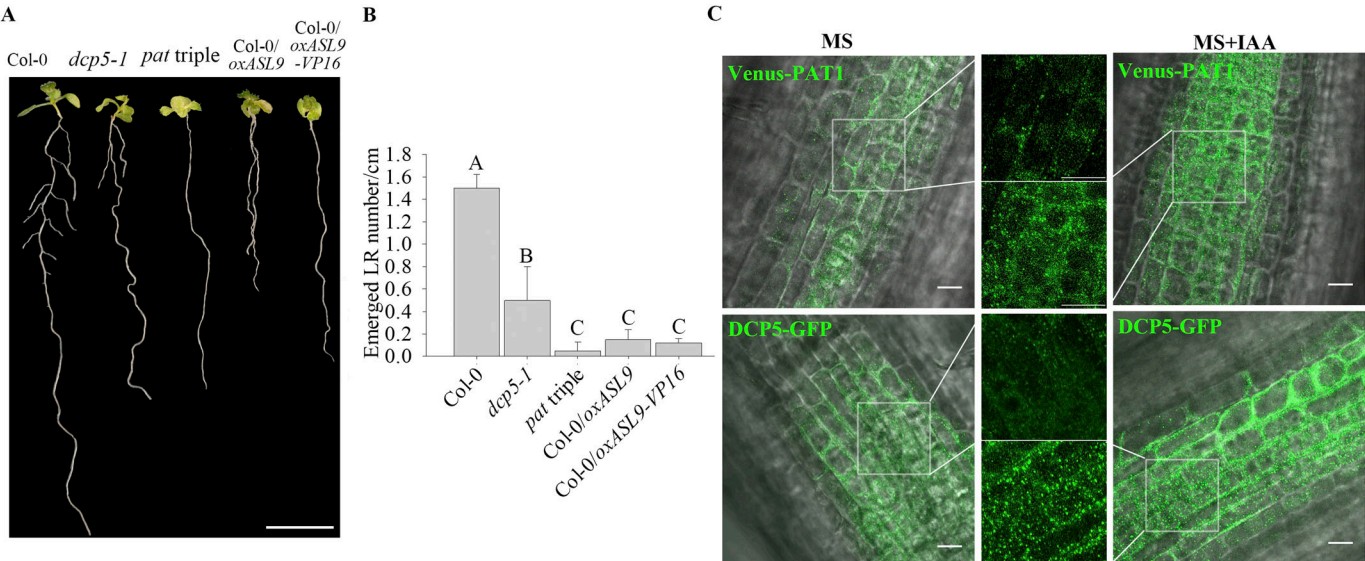

**Figure 3. Accumulation of *ASL9* suppresses LR formation.**
**(A, B)** Phenotypes (A) and emerged LR density (B) of 10-d old seedlings of Col-0, *dcp5-1*, *pat* triple, Col-0/*oxASL9*, and Col-0/*oxASL9-VP16*. The experiment was repeated four times, in each repeat sample size (n)>10 for each genotype, and representative pictures are shown. The scale bar indicates 1 cm. Bars marked with the same letter are not significantly different from each other (*P*-value > 0.05). **(C)** Representative confocal microscopy pictures of root regions from 7-d old seedlings with either Venus–PAT1 or DCP5–GFP treated with MS or MS + 0.2 *μ*M IAA for 15 min. Scale bars indicate 10 *μ*m.

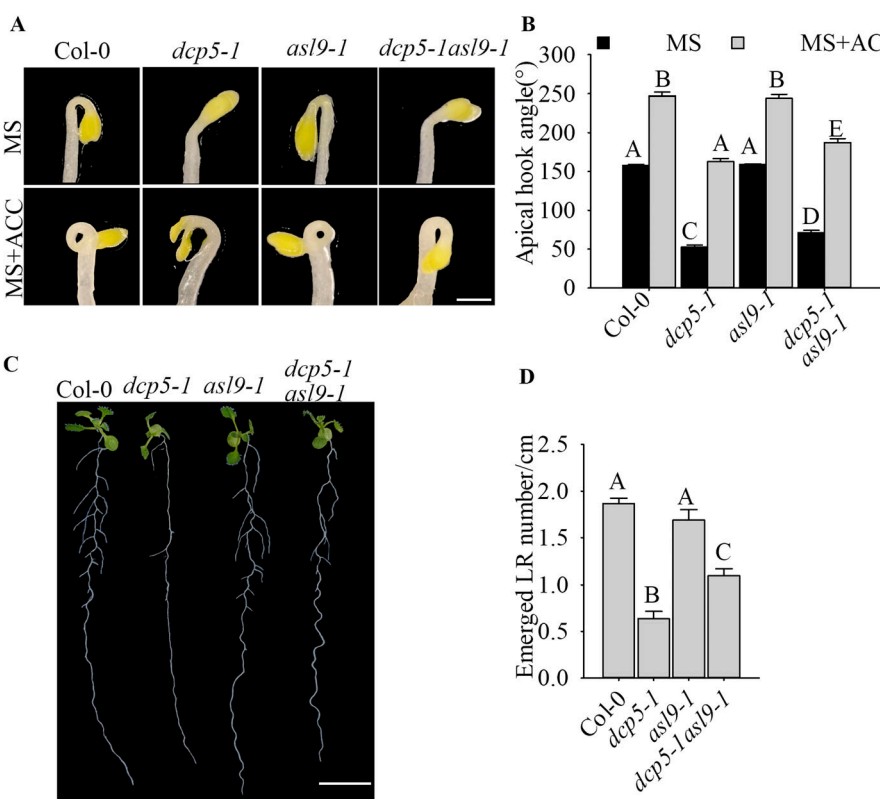

**Figure 4. *ASL9* directly contributes to apical hooking and LR formation.**
**(A, B)** Hook phenotypes (A) and apical hook angles (B) in triple responses to ACC treatment of etiolated Col-0, *dcp5-1*, *asl9-1*, and *dcp5-1asl9-1* seedlings. The treatment was repeated three times, in each repeat sample size (n)>50 for each genotype and treatment, and representative pictures are shown. The scale bar indicates 1 mm. **(C, D)** Phenotypes (C) and emerged LR density (D) of 10-d old seedlings of Col-0, *dcp5-1*, *asl9-1*, and *dcp5-1asl9-1*. Treatment was repeated three times, in each repeat sample size (n) >20 for each genotype, and representative pictures are shown. The scale bar indicates 1 cm. Bars marked with the same letter are not significantly different from each other (*P*-value > 0.05).

hooking and lateral root formation represent classic examples of auxin-dependent development (Peer et al, 2011). In support of this, *axr1* mutants showed defective apical hook formation and reduced LR numbers (Estelle & Somerville, 1987; Lehman et al, 1996). We therefore examined cytokinin- and auxin-related gene expression in both mRNA decay–deficient mutants and *ASL9* over-expressor (Figs S4 and S5). The cytokinin responsive and signaling repressors type-A ARR genes *ARR8* and *ARR15*, the auxin-induced gene *SAUR23* and the auxin biosynthesis gene *TAR2* are all repressed in these genotypes tested, which suggest a misregulation of cytokinin signaling and abrogated auxin homeostasis. To test if the developmental defects of mRNA decay mutants and Col-0/*oxASL9* are due to misregulation of cytokinin, we interfered with cytokinin pathways in *ASL9* over-expressors and decapping mutant *dcp5-1* by knocking out cytokinin-signaling activators *ARR10* and *ARR12* (Ishida et al, 2008). Interestingly, both apical hooking and LR formation phenotypes of *ASL9* over-expressors were largely restored in *arr10-5arr12-1* background (Fig 5), indicating that the developmental defects in *ASL9* over-expressors are most likely caused by misregulation of cytokinin signaling. As for *dcp5-1*, the apical hooking and LR phenotype were partially restored by mutating *arr10* and *arr12* (Fig 6), which despite not reaching the same extend as seen in *ASL9* over-expressors, was still similar to our observations in *dcp5-1asl9-1* double mutants (Figs 4 and 5). Furthermore, the expression of *ARR8*, *ARR15*, *SAUR23*, and *TAR2* in *dcp5-1* was also partially restored in *arr10-5arr12-1* background (Fig S5). Therefore, our data suggest that apical hooking and LR developmental defects in *ASL9* over-expressors and to some degree in mRNA-decapping mutants depend on functional cytokinin signaling.

To test if repressed auxin signaling is also responsible for the developmental defects in mRNA-decapping mutants and *ASL9* over-expressors, we first confirmed the repressed auxin signaling in mRNA decay mutants by introducing the indirect auxin-responsive reporter *DR5*::GFP. We found increased GFP signals in the concave side of Col-0 apical hook region when dark-grown on MS with/without ACC but not in *dcp5-1* or *dcp2-1* under either growth condition, and the overall GFP signals in *dcp2-1* were markedly lower than Col-0 (Fig S6). We also examined DR5::GFP signal in the root area of 7-d old Col-0 and *dcp5-1* seedlings and again, overall GFP signal in *dcp5-1* were strikingly lower than Col-0 (Fig S7). Collectively, these data confirmed our supposition that repressed auxin responses in the mRNA decapping mutants affect apical hook and root developmental processes. Consistent with this notion, exogenous auxin supplementation (0.2 µM IAA) lead to partial restoration of LR formation in *dcp5-1*, *pat* triple, and Col-0/*oxASL9* (Fig S8). Collectively, our findings indicate that misregulation of cytokinin/auxin responses is partially responsible for the developmental defects in the mRNA decay mutants and *ASL9* over-expressors.

## Discussion

Developmental changes require massive overhauls of gene expression (Miyamoto et al, 2015). Apart from unlocking, effectors needed to install a new program, previous states or programs also need to be terminated (Tatapudy et al, 2017; Rodriguez et al, 2020).

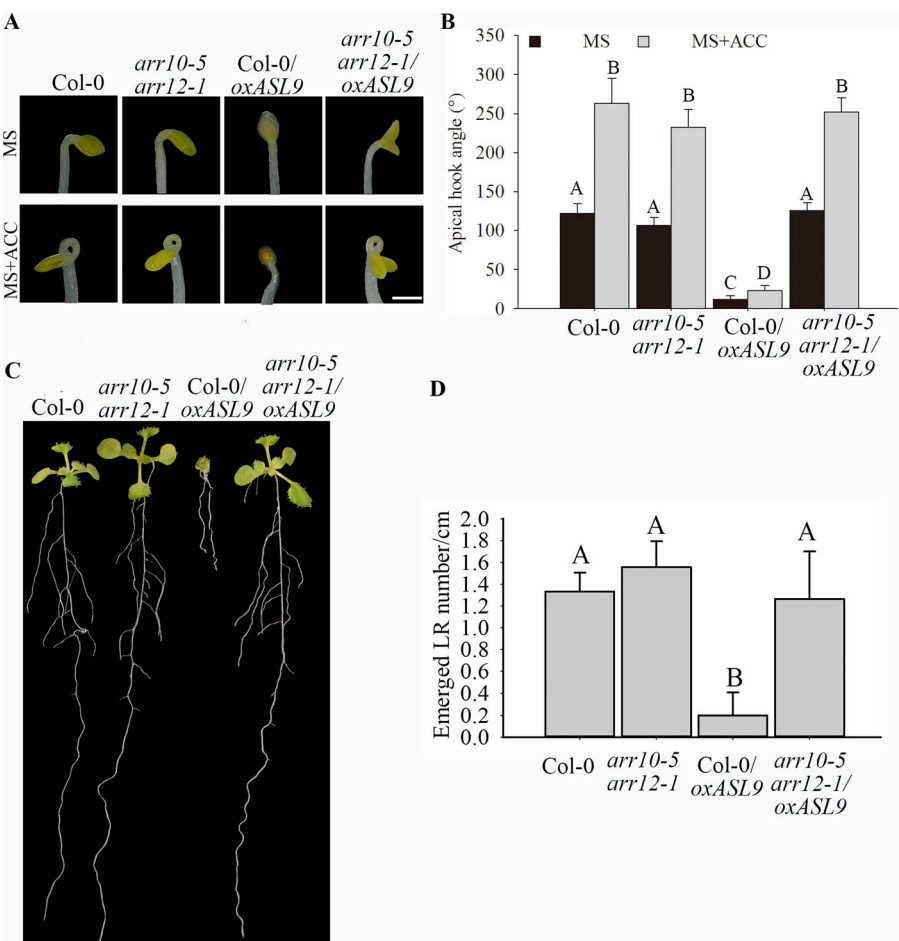

**Figure 5. *ARR10* and *ARR12* loss-of-function restores apical hook and LR formation in *ASL9* over-expressor.**

**(A, B)** Hook phenotypes (A) and apical hook angles (B) in triple responses to ACC treatment of etiolated Col-0, *arr10-5arr12-1*, Col-0/*oxASL9*, and *arr10-5arr12-1/oxASL9* seedlings. The treatment was repeated three times, in each repeat sample size (n)>20 for each genotype and treatment, and representative pictures are shown. The scale bar indicates 1 mm.
**(C, D)** Phenotypes (C) and emerged LR density (D) of 10-d old seedlings of Col-0, *arr10-5arr12-1*, Col-0/*oxASL9*, and *arr10-5arr12-1/oxASL9*. Treatment was repeated three times, in each repeat sample size (n) >10 for each genotype, and representative pictures are shown. The scale bar indicates 1 cm. Bars marked with the same letter are not significantly different from each other (*P*-value > 0.05).

We report here that mRNA decay is required for certain auxin-dependent developmental processes. The stunted growth phenotype and down-regulation of developmental and auxin-responsive mRNAs in the mRNA decapping mutant (Zuo et al, 2022b *Preprint*) supports a model in which defective clearance of mRNAs hampers decision-making upon hormonal perception. Apical hooking and LR formation are classic examples of auxin-dependent developmental processes (Peer et al, 2011). In line with this, we and others observed that mRNA decay–deficient mutants are impaired in apical hooking (Fig 1) and LR formation (Fig 3) (Perea-Resa et al, 2012; Jang et al, 2019). Interestingly, among the transcripts up-regulated in these decay-deficient mutants was that of capped *ASL9/LBD3* (Fig 2), which is involved in cytokinin signaling (Naito et al, 2007). Cytokinin and auxin can act antagonistically (Su et al, 2011), and cytokinin can both attenuate apical hooking (Tantikanjana et al, 2001) and directly affect LR founder cells to prevent initiation of lateral root primordia (Laplaze et al, 2007). Our findings were that defective processing during those developmental events in mRNA decay–deficient mutants involves *ASL9* was supported by our observation that *ASL9* mRNA is directly regulated by the decapping machinery and that Col-0/*oxASL9* transgenic lines cannot reprogram to attain an apical hook or to form LRs (Figs 2 and 3), whereas loss-of-function of *asl9* partially restores the developmental defects in the

decapping deficient mutants (Figs 4 and S3). In line with this, we argue that the misregulation of cytokinin-dependent and auxin-dependent signaling is partially responsible for the developmental defects in mRNA decay–deficient mutants. This is supported by the observation that auxin responses in the *dcp5-1* and *dcp2-1* mutants are repressed (Figs S6 and S7) and treating *dcp5-1*, *pat* triple, and Col-0/*oxASL9* with exogenous auxin partially restores LR formation (Fig S8). Besides misregulation of cytokinin signaling pathway in plants overexpressing *ASL9*, short-term accumulation of *ASL9* also led to down-regulation of cytokinin-responsive genes (Ye et al, 2021), indicating a negative role of *ASL9* in regulating cytokinin responses. However, the fact that the developmental defects in *ASL9* over-expressors are largely restored by knocking out two cytokinin signaling activator genes *ARR10* and *ARR12* suggest the function of *ASL9* during apical hooking, and LR formation largely depends on *ARR10* and *ARR12*. In line with this, the developmental defects of *dcp5-1* are also partially restored in *asl9-1* and *arr10-5arr12-1* backgrounds (Figs 4 and 6), but in addition to *ASL9* and *ARR*s, other unidentified factors also contribute to the defects in apical hook and LR formation in decapping mutants.

Arabidopsis contains 42 *LBD/ASL* genes (Matsumura et al, 2009), among these genes, *LBD16*, *LBD17*, *LBD18*, and *LBD29* control lateral roots formation and regulate plant regeneration (Fan et al, 2012),

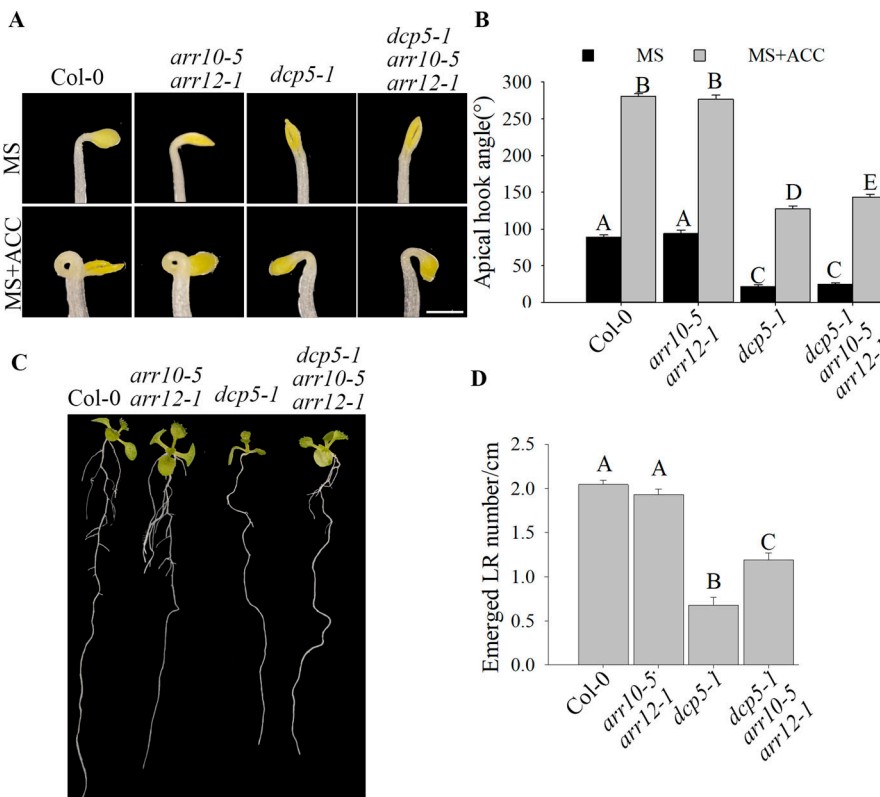

**Figure 6.** ***ARR10* and *ARR12* loss-of-function partially restores apical hook and LR formation in *dcp5-1*.**
**(A, B)** Hook phenotypes (A) and apical hook angles (B) in triple responses to ACC treatment of etiolated Col-0, *arr10-5arr12-1*, *dcp5-1*, and *dcp5-1arr10-5arr12-1* seedlings. The treatment was repeated three times, in each repeat sample size (n)>50 for each genotype and treatment, and representative pictures are shown. The scale bar indicates 1 mm. **(C, D)** Phenotypes (C) and emerged LR density (D) of 10-d old seedlings of Col-0, *arr10-5arr12-1*, *dcp5-1*, and *dcp5-1arr10-5arr12-1*. Treatment was repeated three times, in each repeat sample size (n)>10 for each genotype, and representative pictures are shown. The scale bar indicates 1 cm. Bars marked with the same letter are not significantly different from each other (*P*-value > 0.05).

and overexpression of another member *ASL4* also impairs apical hook (Bell et al, 2012). The partial restoration of apical hooking and LR formation caused by *asl9* mutation in *dcp5-1* and *pat* triple mutant (Figs 4 and S3) suggest that other *ASL*s and/or non-*ASL* genes also contribute to the developmental defects in decapping mutants. Besides lateral root formation, it was recently reported that *Arabidopsis* LBD3, together with LBD4, functions as rate-limiting components in activating and promoting root secondary growth, which is also tightly regulated by auxin and cytokinin, indicating that LBDs balance primary and secondary root growth (Smetana et al, 2019; Smith et al, 2020; Xiao et al, 2020; Ye et al, 2021). Together with auxin, cytokinin plays crucial roles in vascular development through the two-component signaling system, and plants with mutations in cytokinin receptor or type B-*ARRs* exhibit vasculature defects (Mähönen et al, 2006; Yokoyama 2007; Kondo et al, 2011). Hence, we cannot exclude the possibility that the developmental defects we observed in mRNA decapping mutants and *ASL9* over-expressors are also related to their vascular development.

Deadenylated mRNA can be degraded via either 3′–5′ exosomal exonucleases and SOV/DIS3L2 or via the 5′–3′ XRN activity of the decapping complex (Garneau et al, 2007; Sorenson et al, 2018). Sorenson et al (2018) found that *ASL9* expression is dependent on both VCS and SOV based on their transcriptome analysis, so that *ASL9* might be a target of both pathways (Sorenson et al, 2018). Although more direct data are needed to conclude whether SOV can directly regulate *ASL9* mRNA levels, we have shown that *ASL9* is a target of the mRNA decapping machinery. However,

because the Col-0 accession is a *sov* mutant and has no developmental defects, the SOV decay pathway probably only plays an accessory role in regulation of *ASL9*. The function of PBs in mRNA regulation has been controversial because mRNAs in PBs may be sequestered for degradation or re-enter polysomal translation complexes (Franks & Lykke-Andersen, 2008). Yeast PAT1 has also been found to repress translation (Coller & Parker, 2005), and a recent study has confirmed that PBs function as mRNA reservoirs in dark-grown *Arabidopsis* seedlings (Jang et al, 2019). These data open the possibility that *ASL9* might be also regulated at the translational level by the decapping machinery. Nevertheless, our finding of direct interaction of *ASL9* transcripts with DCP5 and PAT1, together with the accumulation of capped *ASL9* in mRNA decay mutants, indicates that *ASL9* misregulation in *dcp2-1*, *dcp5-1*, and *pat* triple mutants is due to mRNA decapping deficiency (Fig 2).

# Materials and Methods

## Plant materials and growth conditions

*Arabidopsis thaliana* ecotype Columbia (Col-0) was used as a control. All mutants used in this study are listed in Table S1. T-DNA insertion lines for AT5G13570 (*DCP2*) *dcp2-1* (Salk_000519), At1g26110 (*DCP5*) *dcp5-1* (Salk_008881), and double mutant *arr10-5arr12-1* have been described (Xu et al, 2006; Ishida et al, 2008; Xu & Chua, 2009). The T-DNA line for AT1g16530 (*ASL9*) is SAIL_659_D08 with

insertion in the first exon. Primers for newly described T-DNA lines are provided in Table S2. *pat* triple mutant, Venus–PAT1, and DCP5–GFP transgenic lines have also been described (Chicois et al, 2018; Zuo et al, 2022b Preprint). The YFP-WAVE line was from NASC (Geldner et al, 2009). Col-0/*oxASL9* line has been described before (Naito et al, 2007).

Plants were grown in 9 × 9 cm or 4 × 5 cm pots at 21°C with 8/16 h light/dark regime, or on plates containing Murashige–Skoog (MS) salts medium (Duchefa), 1% sucrose, and 1% agar with 16/8 h light/dark.

## Plant treatments

For ethylene triple response assays, seeds were plated on normal MS and MS + 50 µM ACC, vernalized 96 h, and placed in the dark at 21°C for 4 d before pictures were taken. Apical hook angle is defined as 180° minus the angle between the tangential of the apical part with the axis of the lower part of the hypocotyl, in the case of hook exaggeration, 180° plus that angle is defined as the angle of hook curvature (Vandenbussche et al, 2010). Cotyledon and hook regions of etiolated seedlings were collected after placing in the dark at 21°C for 4 d for gene expression and XRN1 assay. For LR formation assays, seeds on MS plates were vernalized 96 h and grown with 16/8 h light/dark at 21°C vertically for 10 d. For external IAA application for LR formation experiments, seeds on MS plates were vernalized 96 h and grown with 16/8 h light/dark at 21°C for 7 d and the seedlings were moved to MS or MS+IAA plates and grown vertically for 7 d.

## Cloning and transgenic lines

pGreenIIM DR5V2-ntdtomato/DR5-n3GFP has been published previously (Liao et al, 2015). Arabidopsis transformation was performed by floral dipping (Clough & Bent, 1998) for Col-0/DR5::GFP and thereafter Col-0/DR5::GFP was crossed to *dcp5-1* and *dcp2-1^het* to achieve *dcp5-1*/DR5::GFP and *dcp2-1*/DR5::GFP. *arr10-5arr12-1/oxASL9* was generated by vacuum infiltrating *arr10-5arr12-1* with *A. tumefaciens* strain EHA101 harbouring pSK1-ASL9 (Naito et al, 2007). Transformants were selected on hygromycin (30 mg/l) or methotrexate (0.1 mg/l) MS agar, and survivors were tested for transcript expression by qRT-PCR and protein expression by immuno-blotting and at least two independent lines were used for further analysis.

## Protein extraction, SDS–PAGE, and immunoblotting

Tissue was ground in liquid nitrogen and 4 × SDS buffer (Novex) was added and heated at 95°C for 5 min, cooled to room temperature for 10 min, samples were centrifuged 5 min at 15,682*g*. Supernatants were separated on 10% SDS–PAGE gels, electroblotted to PVDF membrane (GE Healthcare), blocked in 5% (wt/vol) milk in TBS-Tween 20 (0.1%, vol/vol), and incubated 1 h to overnight with primary antibodies (anti-GFP [1:5,000; AMS Biotechnology]). Membranes were washed 3 × 10 min in TBS-T (0.1%) before 1 h incubation in secondary antibodies (anti-rabbit HRP or AP conjugate [1:5,000; Promega]). Chemiluminescent substrate (ECL Plus, Pierce) was applied before camera detection. For AP-conjugated primary antibodies, membranes were incubated in NBT/BCIP (Roche) until bands were visible.

## Confocal microscopy

Imaging was performed using a Zeiss LSM 700 confocal microscope. The confocal images were analyzed with Zen2012 (Zeiss) and ImageJ software. Representative maximum intensity projection images of 10 Z-stacks per image have been shown in Figs 1, 3, S5, and S6.

## RNA extraction and qRT-PCR

Total RNA from tissues was extracted with TRIzol Reagent (Thermo Fisher Scientific), 2 µg total RNA were treated with DNAse I (Thermo Fisher Scientific), and reverse transcribed into cDNA using RevertAid First Strand cDNA Synthesis Kit according to the manufacturer's instructions (Thermo Fisher Scientific). The *ACT2* gene was used as an internal control. qRT-PCR analysis was performed on a Bio-Rad CFX96 system with SYBR Green Master Mix (Thermo Fisher Scientific). Primers are listed in Table S2. All experiments were repeated at least three times each in technical triplicates.

## In vitro XRN1 susceptibility assay

Transcripts XRN1 susceptibility was determined as described (Mukherjee et al, 2012; Kiss et al, 2016) with some modification. Total RNA was extracted from tissues using the NucleoSpin RNA Plant kit (Machery-Nagel). 1 µg RNA was incubated with either 1 unit of XRN1 (New England Biolabs) or water for 2 h at 37°C, loss of ribosomal RNA bands on gel electrophoresis was used to ensure XRN1 efficiency, after heating inactivation under 70°C for 10 min, half of the digest was then reverse transcribed into random primed cDNA with RevertAid First Strand cDNA Synthesis Kit (Thermo Fisher Scientific). Capped target transcript accumulation was measured by comparing transcript levels from XRN1-treated versus mock-treated samples using qRT-PCR (*EIF4A1* serves as inner control) for the individual genotypes (Mukherjee et al, 2012; Roux et al, 2015; Kiss et al, 2016).

## RIP assay

RIP was performed as previously described (Streitner et al, 2012). 1.5 g tissues were fixated by vacuum infiltration with 1% formaldehyde for 20 min followed by 125 mM glycine for 5 min. Tissues were ground in liquid nitrogen and RIP lysis buffer (50 mM Tris–HCl, pH 7.5; 150 mM NaCl; 4 mM MgCl2; 0.1% Igepal; 5 mM DTT; 100 U/ml Ribolock [Thermo Fisher Scientific]; 1 mM PMSF; protease inhibitor cocktail [Roche]) was added at 1.5 ml/g tissue powder. Following 15 min centrifugation at 4°C and 15,682*g*, supernatants were incubated with GFP-Trap A beads (ChromoTek) for 4 h at 4°C. Beads were washed three times with buffer (50 mM Tris–HCl, pH 7.5; 500 mM NaCl; 4 mM MgCl2; 0.5% Igepal; 0.5% sodium deoxycholate; 0.1% SDS; 2 M urea; 2 mM DTT before RNA extraction with TRIzol reagent [Thermo Fisher Scientific]). Transcript levels in input and IP samples were quantified by qRT-PCR, and levels in IP samples were corrected with their own input values and then normalized to YFP-WAVE lines for enrichment.

## 5′-RACE assay

5′-RACE assay was performed using the FirstChoice RLM-RACE kit (Thermo Fisher Scientific) following manufacture's instruction. RNAs were extracted from 4-d-old etiolated seedlings with the NucleoSpin RNA Plant kit (Machery-Nagel), and PCRs were performed using a low (26–28) or high (30–32) number of cycles. Specific primers for the 5′ RACE adapter and for the genes tested are listed in Table S2.

## Statistical analysis

Statistical details of experiments are reported in the figures and legends. Systat software was used for data analysis. Statistical significance between groups was determined by one-way ANOVA (analysis of variance) followed by Holm–Sidak test.

# Supplementary Information

# Acknowledgements

We thank Qi-Jun Chen for Phee401, Nam-Hai Chua for *dcp5-1* and *dcp2-1* seeds, and Damien Garcia for DCP5-GFP marker line seeds. Special thanks to John Mundy for advice throughout the project and critically reading the manuscript. This work was supported by the Novo Nordisk Fonden and the Hartmanns Fond to M Petersen (NNF18OC0052967 and A32638), Biotechnology and Biological Sciences Research Council to the John Innes Centre to L Østergaard (BB/P013511/1) and a PhD scholarship from China Scholarship Council to Z Zuo (201504910714). Microscopy was performed at the Center for Advanced Bioimaging, University of Copenhagen.

## Author Contributions

Z Zuo: conceptualization, data curation, software, formal analysis, validation, investigation, visualization, methodology, and writing—original draft, review, and editing.
ME Roux: conceptualization, data curation, and visualization.
JR Chevalier and SD Højgaard: data curation and investigation.
YF Dagdas: data curation, software, and investigation.
T Yamashino: resources, investigation, and visualization.
E Knight: investigation and methodology.
L Østergaard: resources, investigation, and methodology.
E Rodriguez: investigation and writing—review and editing.
M Petersen: conceptualization, resources, data curation, supervision, funding acquisition, validation, investigation, project administration, and writing—review and editing.

## Conflict of Interest Statement

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
