## [Reviewer comments · Life Science Alliance]

Life Science Alliance

The mRNA decapping machinery targets LBD3/ASL9 to mediate apical hook and lateral root development

Zhangli Zuo, Milena Roux, Jonathan Chevalier, Yasin Dagdas, Takafumi Yamashino, Soren Hojgaard, Emilie Knight, Lars Østergaard, Eleazar Rodriguez, and Morten Petersen

DOI: <https://doi.org/10.26508/lsa.202302090>

Corresponding author(s): Morten Petersen, University of Copenhagen

Review Timeline:

Submission Date:	2023-04-11
Editorial Decision:	2023-05-22
Revision Received:	2023-06-08
Editorial Decision:	2023-06-12
Revision Received:	2023-06-15
Accepted:	2023-06-15

Transaction Report:

Please note that the manuscript was reviewed at Review Commons and these reports were taken into account in the decision-making process at *Life Science Alliance*.

Review
COMMONS

Review #1

In this manuscript, the authors describe the role of the mRNA decay machinery in apical hook formation during germination in darkness in *A.thaliana*. As reported, this machinery is predominantly described in literature in stress response processes, whereas little is known about its involvement during developmental processes. In detail, the authors demonstrated, via RNA immunoprecipitation (RIP) and genetic experiments, the direct regulation of the LATERAL ORGAN BOUNDARIES DOMAIN 3 (LBD3)/ASYMMETRIC LEAVES 2-19 LIKE 9 (ASL9) mRNA stability by the mRNA decapping machinery subunits DCPs. According to the manuscript, ASL9 controls apical hooking, LR development and primary root growth this regulating cytokinin signalling and hence its regulation helps to maintain a correct balance of auxin/cytokinin. Indeed, they showed an impair apical hooking and LR defects both in mRNA decapping mutants, where they observed more capped ASL9 compared to WT, and in ASL9 over-expressor lines. Moreover, they reported a largely restoration of over-expressor lines phenotype in the *arr10-5arr12-1* double mutants.

This work present simple but interesting data that corroborate the authors hypothesis. Nonetheless, I have both major comments and minor comments to improve the manuscript:

Major comments

1. I am a bit concerned by the fact that cytokinin, auxin, LBD3,ARR12 and ARR10 have been largely involved in vasculature development and that the obtained results might be due to their role in vasculature development more than in LBD3 mRNA decapping process. Authors should provide evidence that their results are independent from vasculature defects present in those backgrounds or in case discuss this possibility.
2. The interaction between the described players and auxin is not clear. From the reported experiments it is difficult to understand what authors wants to report as in S4 and S5 are reported experiments not fully described in the text (authors report about introgression of DR5::GFP in *dcp1* and *2* mutants, but SD4 reports ACC treatments of DR5::GFP,*dcp2* mutants and SD5 of 7 dpv root meristems of this strain). Please describe and discuss better the experiment. Also, to this reviewer it is difficult to understand whether the absence of auxin activity in the *dcp2* mutants hypocotyl is merely an effect of the lack of the hook formation in this background or a cause. Please clarify this point including new experiments (*axr1* or *axr3* mutants might help in understand this point).
3. Authors conclude that mRNA decapping is also involved in root growth. However, they do not report direct evidences regarding root growth but mostly regarding the mere root length at a precise developmental stage. Please eliminate this point or provide new experiments (e.g., root length and root meristem activity over time)
4. Regarding root growth defects, these might be due to defect in the vasculature development, please analyse this point or report new experiments (e.g., vasculature analysis of *dcp1,2* mutants or tissue specific expression of DCP2).
5. For consistency the last paragraph of result section: "ASL9 directly contributes to apical hooking, LR formation and primary root growth" should be part of the result section entitled "Accumulation of ASL9 suppresses LR formation and primary root growth". Authors should move this result in the paragraph before "Interference of a cytokinin pathway and/or exogenous auxin restores developmental defects of ASL9 over-expressor and mRNA decay deficient mutants".
6. I suggest being consistent in the description of the statistical analysis. In particular:
 - I suggest reporting the meaning of ANOVA letters and the P-value in each figure as sometimes these information are missing, especially in Fig.2.
 - in Fig.S3 please report the statistical significance on bars and the statistical analysis performed.

Minor comments

L31- please replace "normal" with "proper"

L42-please report the acronym of *axr1*

L57, L59-please include the entire name of DCP2 and XRN

- Please report how many plants were analysed in legend or in methods section
- Please report how many transgenic independent lines were obtained in methods section
- Please, try to change the colours of the graph in Fig.S2A-B, as it quite difficult to distinguish light grey shades.
- In Fig. 5A and S5A scale bars are missing.

The manuscript is interesting and analyse important and overlooked aspects of the role of mRNA decapping in development. Nonetheless experiments reported are not particularly innovative and not always sound. Also authors analysis are a bit superficial probably because they decide to utilize too many systems in their research (root development, hook development and lateral root development).

Review #2

Major Comments

1. My main concern is about the authors' conclusions on the role of mRNA decay and ASL9/LBD3 in the control over cytokinin and auxin responses. I don't think that based on the data presented the authors may do the conclusions stated on lines 184-185, see also the points below.
2. The conclusion about the role of ASL9 and its direct involvement in the apical hook formation and lateral root development/main root growth is a bit exaggerated, based on rather tiny effects mediated by the introduction of *asl9-1* into the *dcp5-1*. Rather, the data suggest that misregulation of other transcripts in the mRNA decay-deficient lines might be responsible for the observed defects. That is also apparent from slightly different phenotypes seen in *dcp5-1/pat* triple compared to *oxASL9* (Fig. 3A). The strong dependency of *oxASL9* phenotype on the presence of functional ARR10 and ARR12 implies cytokinin signaling-dependent mechanism of ASL9/LBD3 action (see also point 3 below). Considering the aforementioned phenotype differences between the *dcp5-1/pat* triple and *oxASL9*, it would be interesting to see the possible dependence of the mRNA decay-deficient line phenotypes on the cytokinin signaling, too.
3. Also the hypothesis on the upregulation of cytokinin signaling in the mRNA decay mutants and *Col-0/oxASL9* is very indirect and should be tested using e.g. TCSn:GFP. The type A ARRs (RRAs) are not only the negative regulators of cytokinin signaling, but also the cytokinin primary response genes. Thus, the downregulation of RRAs could mean the downregulation of the cytokinin signaling pathway in the mRNA decay mutants and/or *Col-0/oxASL9*. The latter seems to be the case as shown recently (Ye et al., 2021).
4. The hypothesis on the causal link between the observed auxin-related defects and upregulated cytokinin signaling (Discussion, lines 214-216) is more than speculation. This could be tested by introducing *arr10 arr12* into the *dcp2-1/DR5-GFP* and/or *dcp5-1/DR5-GFP*.
5. Compared to the text/quantification of the effect of *asl9-1* mutant on the hook formation (Fig. S1D), I see exaggerated hook formation both in the presence and absence of ACC in *asl9-1*, at least on the figures shown in Fig. S1C. Are the shown seedlings not representative?

Minor Comments

1. Syntax problem in the sentence on lines 45-46 (?).
2. The sentence on lines 48-49 should be rephrased. It implies the cytokinins regulate the amount of RRBs, which is not correct (cytokinins control phosphorylation of RRBs, not their abundance, RRAs are not TFs).
3. In the FL for Fig. 2F there is mentioned that MYC-YFP was used as a control compared to the main text mentioning YFP-WAVE (?).
4. Naito et al. (2007) suggest ASL9 as a target of cytokinin signaling, but I don't think they imply the involvement of ASL9 in the cytokinin signaling as mentioned e.g. on line 166 (?).

References

Ye L, Wang X, Lyu M, Siligato R, Eswaran G, Vainio L, Blomster T, Zhang J, Mahonen AP. 2021. Cytokinins initiate secondary growth in the Arabidopsis root through a set of LBD genes. *Curr Biol* 31(15): 3365-3373 e3367.

The authors provide interesting data suggesting possible role of mRNA decay machinery in the hook and lateral root formation and main root growth via decapping-mediated control over ASL9/LBD3 transcript abundance. Based on the observed interaction of the observed phenotypes with hormonal regulations, the authors' conclude mechanistic link between the mRNA decay/ASL9 and cytokinin and auxin responses.

Corresponding author(s): Morten, Petersen

1. General Statements

[This section is optional. Insert here any general statements you wish to make about the goal of the study or about the reviews.]

Thank you for conducting the peer-review of our manuscript. We really appreciate the constructive criticism of the reviewers. However, both reviewers note we over-interpretate our hypothesis about the function of *ASL9* in cytokinin and auxin responses which is not always supported by our data (see also cover letter). Although we agree with those considerations, we feel they are a little forgetful of crediting the other discoveries. Thus, based on their feedback, we have toned down our claims and performed additional experiments and analyses and addressed all the comments raised by both reviewers. We hope this substantially revised and improved version of our manuscript will be better accepted.

Reviewer #1 (Evidence, reproducibility and clarity (Required)):

In this manuscript, the authors describe the role of the mRNA decay machinery in apical hook formation during germination in darkness in *A.thaliana*. As reported, this machinery is predominantly described in literature in stress response processes, whereas little is known about its involvement during developmental processes. In detail, the authors demonstrated, via RNA immunoprecipitation (RIP) and genetic experiments, the direct regulation of the LATERAL ORGAN BOUNDARIES DOMAIN 3 (LBD3)/ASYMMETRIC LEAVES 2-19 LIKE 9 (ASL9) mRNA stability by the mRNA decapping machinery subunits DCPs. According to the manuscript, ASL9 controls apical hooking, LR development and primary root growth is regulating cytokinin signalling and hence its regulation helps to maintain a correct balance of auxin/cytokinin. Indeed, they showed an impair apical hooking and LR defects both in mRNA decapping mutants, where they observed more capped ASL9 compared to WT, and in ASL9 over-expressor lines. Moreover, they reported a largely restoration of over-expressor lines phenotype in the *arr10-5arr12-1* double mutants.

This work present simple but interesting data that corroborate the authors hypothesis.

Our response: We thank the reviewer for acknowledging the significance of our findings although we wonder what it's meant by "simple data". Through a combination of (complicated) genetics, phenotyping, cell imaging and molecular biology, we have provided mechanistic evidence on the function of the decapping machinery during 2 different post embryonic

developmental events. Please see our detailed answers to the reviewer's comments in the following.

Nonetheless, I have both major comments and minor comments to improve the manuscript:
MAJOR COMMENTS:

1. I am a bit concerned by the fact that cytokinin, auxin, LBD3, ARR12 and ARR10 have been largely involved in vasculature development and that the obtained results might be due to their role in vasculature development more than in LBD3 mRNA decapping process. Authors should provide evidence that their results are independent from vasculature defects present in those backgrounds or in case discuss this possibility.

Our response: We are a bit puzzled on how vasculature development could explain the apical hook phenotype observed in the decapping mutant. Data like the rapid assembly of P-bodies upon IAA (Fig. 3C) treatments and the overall decreased DR5 signal in *dcp* mutants (Fig.S5&6) are all consistent with a process precluding vasculature formation. However, we still discuss the possibility that the developmental defects observed in mRNA decapping mutants and *ASL9* overexpressor might be related to the vasculature development in these plants (Line 239-244).

2. The interaction between the described players and auxin is not clear. From the reported experiments it is difficult to understand what authors wants to report as in S4 and S5 are reported experiments not fully described in the text (authors report about introgression of DR5::GFP in *dcp1* and 2 mutants, but SD4 reports ACC treatments of DR5::GFP, *dcp2* mutants and SD5 of 7 dpg root meristems of this strain). Please describe and discuss better the experiment. Also, to this reviewer it is difficult to understand whether the absence of auxin activity in the *dcp2* mutants hypocotyl is merely an effect of the lack of the hook formation in this background or a cause. Please clarify this point including new experiments (*axr1* or *axr3* mutants might help in understand this point).

Our response: We follow the reviewer's suggestions and trust we now describe and discuss Fig S5&6 (old Fig S4&S5) clear in Line 188-193. As *axr1* has been published with apical hook and lateral root defect (old Line 42, new Line 39&169), we did not repeat it in new experiments but emphasize it in Line 169.

3. Authors conclude that mRNA decapping is also involved in root growth. However, they do not report direct evidences regarding root growth but mostly regarding the mere root lenght at a precise developmental stage. Please eliminate this point or provide new experiments (e.g., root length and root meristem activity over time)

Our response: We follow the reviewer's suggestions and eliminate the data regarding to primary root growth (Fig. 3-6 &S2)

4. Regarding root growth defects, these might be due to defect in the vasculature development, please analyse this point or report new experiments (e.g., vasculature analysis of *dcp1,2* mutants or tissue specific expression of DCP2).

Our response: We largely agree with the reviewer, all the decapping components DCP1, DCP2, DCP5 and PAT1 exhibit high expression in xylem cells and low expression in procambium cells (Brady et al., 2007) indicating functions of decapping components in vasculature development. However, we did not include this knowledge in our manuscript since we decided to eliminate the primary root growth data (Fig.3-6&S2).

5. For consistency the last paragraph of result section: "ASL9 directly contributes to apical hooking, LR formation and primary root growth" should be part of the result section entitled "Accumulation of ASL9 suppresses LR formation and primary root growth". Authors should move this result in the paragraph before "Interference of a cytokinin pathway and/or exogenous auxin restores developmental defects of ASL9 over-expressor and mRNA decay deficient mutants".

Our response: We agree thus we reorganize the result sections and move "ASL9 directly contributes to apical hooking and LR formation" before "Interference of a cytokinin pathway and/or exogenous auxin restores developmental defects of ASL9 over-expressor and mRNA decay deficient mutants" (Line 152).

6. I suggest being consistent in the description of the statistical analysis. In particular:
- I suggest reporting the meaning of ANOVA letters and the P-value in each figure as sometimes these information are missing, especially in Fig.2.

Our response: We used ANOVA letters when comparing among genotypes and treatments, for example Fig 2A; and we used stars when comparing to controls, for example old Fig 2F. For consistency, we use letters for all the statistical analysis now and we report the meaning of the letters clearly in the figure legends (Fig. 1-6, S1-5&7). However, we think that putting the P-values in each figure would not be reader-friendly, and thus we have not done this.

- in Fig.S3 please report the statistical significance on bars and the statistical analysis performed.

Our response: We thank the reviewer for pointing it out, we report the statistical analysis now in new Fig. S2 (old Fig. S3).

MINOR COMMENTS:

L31- please replace "normal" with "proper"

Our response: We thank the reviewer for the suggestion, now we replace "normal" with "proper"(Line 30)

L42-please report the acronym of *axr1*

Our response: The acronym of *axr1* is correctly reported (Line 40).

L57, L59-please include the entire name of DCP2 and XRN

Our response: The entire name of DCP2 and XRN are correctly included (Line 55 &57).

-Please report how many plants were analysed in legend or in methods section

Our response: The numbers of plants in analysis are now reported in figure legends (Fig. 1-6, S1,2&7).

-Please report how many transgenic independent lines were obtained in methods section

Our response: The numbers of transgenic independent lines are now reported in methods (Line 292)

- Please, try to change the colours of the graph in Fig.S2A-B, as it quite difficult to distinguish light grey shades.

Our response: We thank the reviewer's suggestions, the colours of new Fig.S3&4 (old Fig.S2) are changed.

- In Fig. 5A and S5A scale bars are missing.

Our response: We thank the reviewer for pointing this out, scale bars are correctly added in new Fig 4 & S6 (old Fig 5 & S5).

Reviewer #1 (Significance (Required)):

The manuscript is interesting and analyse important and overlooked aspects of the role of mRNA decapping in development. Nonetheless experiments reported are not particularly innovative and not always sound. Also authors analysis are a bit superficial probably because they decide to utilize too many systems in their research (root development, hook development and lateral root development).

Our response: We thank the reviewer again for acknowledging the significance of our findings and hope we satisfied the reviewer with our answers above. However, we would like to ask what is the purpose of writing “experiments are not particularly innovative”? We admit we used established and robust experiments which we found sufficient to answer the overlooked aspects of the role of mRNA decapping in apical hook and lateral root development as also noted by the reviewer, but maybe we simply don't understand the comment.

Reviewer #2 (Evidence, reproducibility and clarity (Required)):

Major Comments

1. My main concern is about the authors' conclusions on the role of mRNA decay and ASL9/LBD3 in the control over cytokinin and auxin responses. I don't think that based on the data presented the authors may do the conclusions stated on lines 184-185, see also the points below.

Our response: We agree thus we tone down our conclusion in our new manuscript (Line 197-199), see answers below for detail.

2. The conclusion about the role of ASL9 and its direct involvement in the apical hook formation and lateral root development/main root growth is a bit exaggerated, based on rather tiny effects mediated by the introduction of *asl9-1* into the *dcp5-1*. Rather, the data suggest that misregulation of other transcripts in the mRNA decay-deficient lines might be responsible for the observed defects. That is also apparent from slightly different phenotypes seen in *dcp5-1/pat* triple compared to *oxASL9* (Fig. 3A). The strong dependency of *oxASL9* phenotype on the presence of functional ARR10 and ARR12 implies cytokinin signaling-dependent mechanism of ASL9/LBD3 action (see also point 3 below). Considering the aforementioned phenotype differences between the *dcp5-1/pat* triple and *oxASL9*, it would be interesting to see the possible dependence of the mRNA decay-deficient line phenotypes on the cytokinin signaling, too.

Our response: We note restoration of *dcp5-1* developmental defects in *asl9* backgrounds is partial, indicating other ASLs or non-ASLs also contributing to apical hook and lateral root development (old Line 224-225, new Line 229-230 & 234-235). We also note that partial suppression is a common phenomenon when studying discrete developmental traits. Two such examples could include the knockout of *TPXL5* which partially suppressed the increase of LR density in the *hy5* mutant and the introduction of a point mutation in SnRK2.6 in the *gsnor1-3/ost1-3* double-mutant partially suppressed the effect of *gsnor1-3* on ABA-induced stomatal

closure (Qian et al., 2022 *The Plant Cell* doi.org/10.1093/plcell/koac358; Wang et al., 2015 *PNAS* 112, 613). In addition to such discrete developmental traits, more dramatic phenotypes like autoimmunity may also only be partially suppressed (Zhang et al., 2012 *CH&M* 11, 253). However, we agree that it's interesting to check the dependence of cytokinin signaling of the developmental defects in mRNA decay-deficient mutants. Unfortunately, we were only able to cross *arr10 arr12* into *dcp5-1*. This line showed similar partial restoration of *dcp5-1* developmental defects as seen for *dcp5-1asl9-1*. Overall, these data indicates that contribution of mRNA decapping targeting *ASL9* transcripts during apical hook and LR formation depends on *ARR10* and *ARR12* (Fig. 4&6, Line 180-186).

3. Also the hypothesis on the upregulation of cytokinin signaling in the mRNA decay mutants and *Col-0/oxASL9* is very indirect and should be tested using e.g. TCSn:GFP. The type A ARRs (RRAs) are not only the negative regulators of cytokinin signaling, but also the cytokinin primary response genes. Thus, the downregulation of RRAs could mean the downregulation of the cytokinin signaling pathway in the mRNA decay mutants and/or *Col-0/oxASL9*. The latter seems to be the case as shown recently (Ye et al., 2021).

Our response: We thank the reviewer for suggesting a different annotation of our result regarding to type-A ARRs. Ye et al reported accumulation of *ASL9/LBD3* induced downregulation of cytokinin pathway based on weaker *ARR5* and TCSn-GFP signal (Ye et al., 2021). However, the fact that knocking out cytokinin signaling activator genes *ARR10* and *ARR12* largely restored developmental defects in *ASL9* over-expressors lead to the hypothesis of upregulated cytokinin signaling in *ASL9* over-expressors (Fig 5). Therefore, we substitute “upregulation” with “misregulation” for cytokinin signaling to compromise in our new manuscript (Line 174).

4. The hypothesis on the causal link between the observed auxin-related defects and upregulated cytokinin signaling (Discussion, lines 214-216) is more than speculation. This could be tested by introducing *arr10 arr12* into the *dcp2-1/DR5-GFP* and/or *dcp5-1/DR5-GFP*.

Our response: We thank the reviewer for the suggestions, due to time and funds management, we decided to check auxin related gene expression in *dcp5-1arr10-5arr12-1* mutants instead of making transgenic plants in triple mutant. The repressed expression of *SAUR23* and *TAR2* in *dcp5-1* is partially restored (Fig. S4), indicating possible repression of auxin signaling caused by upregulated cytokinin signaling. However, for consistency in cytokinin signaling description, we tone down the hypothesis on the link between auxin-related defects and cytokinin signaling (Line 218-220).

5. Compared to the text/quantification of the effect of *asl9-1* mutant on the hook formation (Fig. S1D), I see exaggerated hook formation both in the presence and absence of ACC in *asl9-1*, at least on the figures shown in Fig. S1C. Are the shown seedlings not representative?

Our response: We thank the reviewer for pointing our mistakes out, the shown seedlings are representative but mislabeled and the mistakes are corrected now in our new manuscript (Fig. S1C).

Minor Comments

1. Syntax problem in the sentence on lines 45-46 (?).

Our response: We thank the reviewer for pointing it out, syntax problem of this sentence is solved now in new manuscript (Line 41-44).

2. The sentence on lines 48-49 should be rephrased. It implies the cytokinins regulate the amount of RRBs, which is not correct (cytokinins control phosphorylation of RRBs, not their abundance, RRAs are not TFs).

Our response: We now rephrase the sentence in a correct way (Line 46)

3. In the FL for Fig. 2F there is mentioned that MYC-YFP was used as a control compared to the main text mentioning YFP-WAVE (?).

Our response: We thank the reviewer for pointing this out, the YFP-WAVE line we used is MYC-YFP transgenic plants, we now include this information in our manuscript (Line 136) and for consistency we changed MYC-YFP to YFP-WAVE in Fig. 2F.

4. Naito et al. (2007) suggest ASL9 as a target of cytokinin signaling, but I don't think they imply the involvement of ASL9 in the cytokinin signaling as mentioned e.g. on line 166 (?)

Our response: We largely agree with the reviewer thus we also cite Ye's paper here in our new manuscript (Line 165)

References

Ye L, Wang X, Lyu M, Siligato R, Eswaran G, Vainio L, Blomster T, Zhang J, Mahonen AP. 2021. Cytokinins initiate secondary growth in the Arabidopsis root through a set of LBD genes. *Curr Biol* 31(15): 3365-3373 e3367.

Reviewer #2 (Significance (Required)):

The authors provide interesting data suggesting possible role of mRNA decay machinery in the hook and lateral root formation and main root growth via decapping-mediated control over ASL9/LBD3 transcript abundance. Based on the observed interaction of the observed phenotypes with hormonal regulations, the authors' conclude mechanistic link between the mRNA decay/ASL9 and cytokinin and auxin responses.

Our response: We thank the reviewer for acknowledging the significance of our findings.

May 22, 2023

Re: Life Science Alliance manuscript #LSA-2023-02090

Prof. Morten Petersen
Copenhagen University
Biology
Ole Maaloees Vej 5
Copenhagen 2200
Denmark

Dear Dr. Petersen,

Thank you for submitting your revised manuscript entitled "The mRNA decapping machinery targets LBD3/ASL9 to mediate apical hook and lateral root development in Arabidopsis" to Life Science Alliance. The manuscript has been seen by original reviewer #2 whose comments are appended below. Some important issues remain.

Our general policy is that papers are considered through only one revision cycle; however, I would be open to one additional short round of revision. Please note that I will expect to make a final decision without additional reviewer input upon re-submission. Please send me a proposal via email on how these remaining points will be addressed. I appreciate that the best course of action may be to modify relevant claims made, rather than further experimentation.

Please submit the final revision within one month, along with a letter that includes a point by point response to the remaining reviewer comments.

To upload the revised version of your manuscript, please log in to your account: <https://lsa.msubmit.net/cgi-bin/main.plex>
You will be guided to complete the submission of your revised manuscript and to fill in all necessary information.

B. MANUSCRIPT ORGANIZATION AND FORMATTING:

Sincerely,

Reviewer #2 (Comments to the Authors (Required)):

In the revised version of their manuscript, the authors did not provide additional evidence/explanations to rebut my concerns I raised in the original review, see below.

Major Comments

1. The authors did not respond in a satisfactory way to my main concern on the specificity of the observed transcript accumulation in mRNA decay-deficient lines neither to ASL9, nor to cytokinin and auxin signaling. As also shown by the authors in related works (Zuo, Z. et al., 2022; Zuo, Zhangli et al., 2022), in the PAT deficient lines, number of transcripts of genes involved in plethora of processes are misregulated, leading to highly pleiotropic phenotype changes.
2. The above is well reflected in the very tiny differences in the phenotypes of *dcp5-1* and *dcp5-1 asl9-1*, thus strongly questioning the causal link between the *dcp5-1* and ASL9 accumulation.
3. Similarly, the partial recovery of SAUR23 and TAR2 levels in *dcp5-1 arr10-5 arr12-1* compared to *dcp5-1* is very weak, again questioning the specificity of observed effects.

Minor Comments

1. Missing reference to Laplaze et al. (2007) on line 45.
2. Causality of the statement on the lines 45-6 ("Thus, reshaping the levels of certain genes leads to changes in cellular identity.") and its link to the preceding text is not clear to me.

References

- Laplaze L, Benkova E, Casimiro I, Maes L, Vanneste S, Swarup R, Weijers D, Calvo V, Parizot B, Herrera-Rodriguez MB, et al. 2007. Cytokinins act directly on lateral root founder cells to inhibit root initiation. *Plant Cell* 19(12): 3889-3900.
- Zuo Z, Roux M, Rodriguez E, Petersen M. 2022. mRNA Decapping Factors LSM1 and PAT Paralogs Are Involved in Turnip Mosaic Virus Viral Infection. *Molecular plant-microbe interactions* : MPMI 35(2): 125-130.
- Zuo Z, Roux ME, Dagdas YF, Rodriguez E, Petersen M. 2022. PAT mRNA decapping factors function specifically and redundantly during development in *Arabidopsis*. *bioRxiv*: 2022.2007.2006.498930.

Dear Editor,

Thanks for the opportunity of resubmitting a revised version of our manuscript (LSA-2023-02090) entitled “The mRNA decapping machinery targets *LBD3/ASL9* to mediate apical hook and lateral root development in *Arabidopsis*”. We have carefully responded to all the comments from the reviewer to our manuscript and thus have decided to resubmit our revised manuscript to *Life Science Alliance*.

We note Reviewer 2 was not convinced by the partial suppression of the *dcp5-1* phenotype when mutating *ASL9*. We still feel the reviewer neglect our work of identifying *ASL9* as a target gene of the mRNA decay pathway. In addition, partial suppression is not an uncommon theme in genetic research when studying complicated traits and we also provided some examples from the literature in our previous rebuttal letter. Nevertheless, to substantiate this notion, we also now demonstrate that mutations in *ASL9* also partially suppress the lack of apical hook and lateral root formation in *pat* triple mutants. To us, this strongly support our conclusion that *ASL9* contributes to some but not all developmental defects seen in decapping mutants. We would also like to emphasize that we agree with the reviewer and find decapping mutants pleiotropic. Many genes probably contribute to these defects in apical hook and lateral root development including *ASL9*. Thus, we do not in any way exclude more genes are mis-regulated in the decapping mutants. This is why we use the terminology “contribute” and our new genetic data further support this conclusion.

In summary, we think it's fair to leave the evaluation of the “partial recovery” to the readers and hope this revised and improved version of our manuscript is acceptable for publication in *Life Science Alliances*.

Reviewer #2 (Comments to the Authors (Required)):

In the revised version of their manuscript, the authors did not provide additional evidence/explanations to rebut my concerns I raised in the original review, see below.

Our response: We are honestly a bit surprised by “did not provide additional evidence/explanations”. We managed to provide solid genetic data on *dcp5-larr10-5arr12-1* in our 1st revised manuscript based on the reviewer's comments (Fig 6) which was not an easy task. Please see our detailed answers to the reviewer's comments in the following.

Major Comments

1. The authors did not respond in a satisfactory way to my main concern on the specificity of the observed transcript accumulation in mRNA decay-deficient lines neither to *ASL9*, nor to cytokinin and auxin signaling. As also shown by the authors in related works (Zuo, Z. et al., 2022; Zuo,

Zhangli et al., 2022), in the PAT deficient lines, number of transcripts of genes involved in plethora of processes are misregulated, leading to highly pleiotropic phenotype changes.

Our response: We thank the reviewer for noticing our other works. However, we did not quite understand the reviewer's comments. We have solid data showing that mRNA decay machinery targets *ASL9* transcripts specifically and *ASL9* accumulate in 3 mRNA decay deficient mutants *dcp2-1*, *dcp5-1* and *pat* triple mutants (Fig 1&2). Furthermore, mutating *ASL9* or cytokinin signaling activator genes *ARR10* and *ARR12*, and exogenously auxin application could all partially restore the developmental defects of mRNA decapping mutants (Fig 4-6,S3&S8). Collectively, all these data support that mRNA decay target *ASL9* and somehow mess with cytokinin and auxin signaling. We agree with the reviewer of course, that mRNA decay mutants have pleiotropic phenotypes (Line 78), thus we do not in any way exclude other genes are also involved in the developmental defects seen in mRNA decapping mutants (Line 234).

2. The above is well reflected in the very tiny differences in the phenotypes of *dcp5-1* and *dcp5-1 asl9-1*, thus strongly questioning the causal link between the *dcp5-1* and *ASL9* accumulation.

Our response: To substantiate our findings we now also show partial restoration of the developmental defects by mutating *ASL9* in *pat* triple mutants (new Fig S3). This strongly indicates other *ASLs* or non-*ASLs* also contribute to such developmental traits, and we think we have emphasized this very clearly in the discussion (Line 234).

3. Similarly, the partial recovery of *SAUR23* and *TAR2* levels in *dcp5-1 arr10-5 arr12-1* compared to *dcp5-1* is very weak, again questioning the specificity of observed effects.

Our response: It's unfortunate that Reviewer 2 did not like the partial recovery of mutating *ASL9* and *ARR10* *ARR12* in the decapping mutants. Again, we agree that one single gene-*ASL9* cannot explain the pleiotropic phenotype of mRNA decay mutants and more genes are involved but also all are replies above.

Minor Comments

1. Missing reference to Laplaze et al. (2007) on line 45.

Our response: We now have the reference for line 45.

2. Causality of the statement on the lines 45-6 ("Thus, reshaping the levels of certain genes leads to changes in cellular identity.") and its link to the preceding text is not clear to me.

Our response: We believe this statement and its link to the preceding text is fine since other reviewers did not question it.

References

Laplaze L, Benkova E, Casimiro I, Maes L, Vanneste S, Swarup R, Weijers D, Calvo V, Parizot B, Herrera-Rodriguez MB, et al. 2007. Cytokinins act directly on lateral root founder cells to inhibit root initiation. *Plant Cell* 19(12): 3889-3900.

Zuo Z, Roux M, Rodriguez E, Petersen M. 2022. mRNA Decapping Factors LSM1 and PAT Paralogs Are Involved in Turnip Mosaic Virus Viral Infection. *Molecular plant-microbe interactions* : MPMI 35(2): 125-130.

Zuo Z, Roux ME, Dagdas YF, Rodriguez E, Petersen M. 2022. PAT mRNA decapping factors function specifically and redundantly during development in *Arabidopsis*. bioRxiv: 2022.2007.2006.498930.

June 12, 2023

RE: Life Science Alliance Manuscript #LSA-2023-02090R

Prof. Morten Petersen
University of Copenhagen
Biology
Ole Maaloees Vej 5
Copenhagen 2200
Denmark

Dear Dr. Petersen,

Thank you for submitting your revised manuscript entitled "The mRNA decapping machinery targets LBD3/ASL9 to mediate apical hook and lateral root development". We would be happy to publish your paper in Life Science Alliance pending final revisions necessary to meet our formatting guidelines.

- please upload your Tables in editable .doc or excel format
- please be aware that the titles on the manuscript file and in the system must match
- before submitting your manuscript, kindly refer to our guidelines for manuscript preparation <https://www.life-science-alliance.org/manuscript-prep>. Please ensure that all sections of your manuscript are accurately labeled and arranged in the correct order.
- please add your main, supplementary figure, and table legends to the main manuscript text after the references section
- please use the [10 author names et al.] format in your references (i.e., limit the author names to the first 10)
- please add callouts for Figures 5A-D; 6A-D; S4B; S5A-B; S8A-F to your main manuscript text

A. FINAL FILES:

B. MANUSCRIPT ORGANIZATION AND FORMATTING:

Sincerely,

June 15, 2023

RE: Life Science Alliance Manuscript #LSA-2023-02090RR

Prof. Morten Petersen
University of Copenhagen
Biology
Ole Maaloees Vej 5
Copenhagen 2200
Denmark

Dear Dr. Petersen,

Thank you for submitting your Research Article entitled "The mRNA decapping machinery targets LBD3/ASL9 to mediate apical hook and lateral root development". It is a pleasure to let you know that your manuscript is now accepted for publication in Life Science Alliance. Congratulations on this interesting work.

DISTRIBUTION OF MATERIALS:

Again, congratulations on a very nice paper. I hope you found the review process to be constructive and are pleased with how the manuscript was handled editorially. We look forward to future exciting submissions from your lab.

Sincerely,
